# Pivotal Prompt Tuning for Video Dynamic Editing

**Input Video**

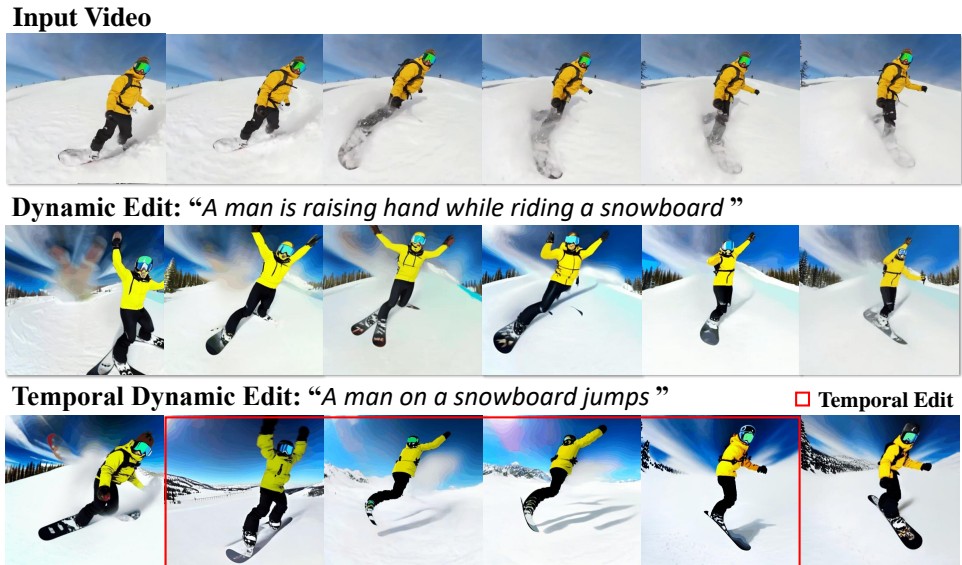

**Dynamic Edit:** "*A man is raising hand while riding a snowboard* "

**Temporal Dynamic Edit:** "*A man on a snowboard jumps* "  □ **Temporal Edit**

Figure 1: PDEdit – Pivotal Dynamic Editing. PDEdit can perform various text-based non-rigid video editing involving large dynamic motion variations and temporal changes while preserving consistency among the neighboring frames.

## Abstract

Text-conditioned image editing has recently provided high-quality edits on images based on diffusion frameworks. Unfortunately, this success did not carry over to video editing, which continues to be challenging. Video editing is limited to rigid editing such as object overlay and style transfer. This paper proposes pivotal dynamic editing (PDEdit) for performing spatial-temporal non-rigid video editing based only on the target text, which has never been attempted before. PDEdit is capable of synthesizing a new pose of an object/person in the video, either at a specific moment or throughout the video, while preserving the temporal consistency of edited motions and a high level of fidelity to the original input video. In contrast to previous works, the proposed method performs editing based only on the input video and target text. It does not require any other auxiliary inputs (e.g., object masks or source video captions). Based on the video diffusion model, PDEdit using the proposed prompt pivoting leverages the target text prompt for editing the input video. The quality and adaptability of the proposed method on numerous input videos from different domains show the proposed to be highly effective. It can produce high-fidelity video edits under a single unified PDEdit framework. The code for this work will be made publicly available.

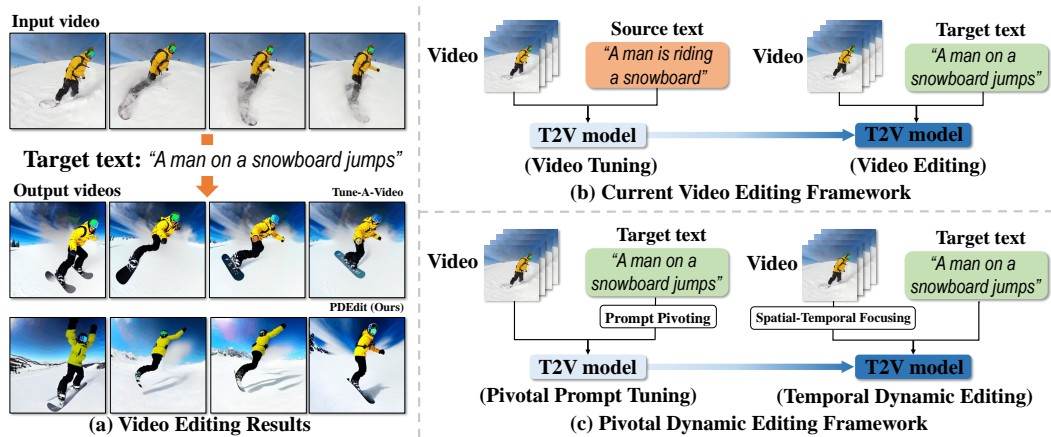

Figure 2: Edited results (a) of existing model and PDEdit on the same video according to the target text. Previous models are not able to perform non-rigid edit (*e.g.*, motion change). Illustrations of video editing frameworks about the current system (b) and our proposed system (c).

# 1 INTRODUCTION

The recent success of generative frameworks (Creswell et al., 2018; Kingma & Welling, 2013; Ho et al., 2020) and large-scale models (Radford et al., 2018; Devlin et al., 2018; Radford et al., 2021) has paved the way for machines to produce surreal outputs that surpass human capability. The diffusion frameworks (Dhariwal & Nichol, 2021; Song et al., 2020c) underpin a building block of such innovative changes, and based on this, many large-scale generative models (Rombach et al., 2022; Song et al., 2020a) have emerged. To be specific, diffusion-based text-to-image (T2I) models (Nichol et al., 2021; Rombach et al., 2022) synthesize high-quality images and further edit (Ruiz et al., 2022; Kawar et al., 2022) the image by freely changing specific attributes corresponding to the input target text while maintaining the others in the image. Extending from the image, diffusion-based text-to-video (T2V) models (Bar-Tal et al., 2022; Wu et al., 2022b) also have been considered. To overcome the insufficient training resources of the video, several respectful technical contributions (Singer et al., 2022; Hong et al., 2022) have been made to distill knowledge into the T2V model from large-scale pre-trained T2I models. Researchers are currently pushing the boundary of text-based video generation to a more controlled and fine-grained approach by generating attributes corresponding to users' needs in the text onto videos, ultimately performing text-based video editing.

In a formal definition of text-based video editing, as shown in Figure 2 (a), systems are given an input video and a target text prompt that describes the desired attributes within the video, where they produce an edited video that conforms to the target text prompt. To perform this, the editing systems largely follow two canonical processes: (1) video tuning and (2) video editing. In video tuning, the system is trained to generate the input video and understand the contextual meaning of the video. In video editing, the system generates the variants of the input video that obey the meaning of the target text prompt. To generate proper variants of video aligned with the target text, pre-trained vision-language models (Radford et al., 2021; Rombach et al., 2022) have been introduced to provide the required attributes for editing. Although recent video editing models (Liu et al., 2023; Ceylan et al., 2023; Qi et al., 2023; Shin et al., 2023) have presented proper capabilities to modify existing frames according to the input text, they are still limited to rigid edits within the scope of inpainting, such as style transfer and object overlay. To be specific, as shown in Figure 2 (a), for the given target text prompt (*e.g.*, "A man on a snowboard jumps") that requires editing of dynamic motion change (*i.e.*, synthesizing a new pose), current video editing systems do not conform to the target prompt and return the original input video under over-fidelity. Otherwise, they often perform impractical edits by keeping a single moment of the changed motion throughout the video, which does not correctly represent the intended complete motion. Therefore the results[1] concerning to complex non-rigid editing (*e.g.*, dynamic and temporal change) are still unsatisfactory.

---

[1]Our categorical analysis of current editing results is further presented in Appendix B.

One of the reasons for the unsatisfactory edits is rooted in the video tuning process of current video editing frameworks. As shown in Figure 2 (b), current editing frameworks require captions (*i.e.*, source text prompt) that describe the video as additional inputs for tuning a video. After tuning, they perform an edit to the video with the target text prompt. However, the use of source prompts causes functionally unnecessary tuning of the video on content (*e.g.*, focusing on the 'riding' by source prompt) unrelated to the required editing (*e.g.*, 'jump'), resulting in suboptimal edits. Furthermore, the results become vulnerable to the variants of the source prompts. Therefore, the frameworks that tune a video by utilizing a source prompt are not suitable for effective text-based video editing.

To this end, we propose pivotal dynamic editing (PDEdit) framework that performs spatial-temporal non-rigid edits to a video with only a target text prompt. As shown in Figure 2 (c), the proposed PDEdit framework includes a pivotal prompt tuning (PPT) that enables the system to tune a video with a target text prompt via our proposed prompt pivoting. The prompt pivoting makes the video tuning independent of the source prompt by providing a textual prompt from the target prompt that aligns with the input video. Furthermore, to provide effective non-rigid editing with respect to dynamic motion changes, PDEdit also includes temporal dynamic editing (TDE) which promotes motion changes in spatial and temporal domains by maximizing the effect of the edit in the intended moment via our proposed spatial-temporal focusing. PDEdit enhances the overall visual appeal of the video and ensures that edits are seamlessly integrated into the original footage. Therefore, PDEdit pursues the general format of text-based video editing and performs various types of editing including style transfer, object overlay, and motion changes. Extensive experiments on PDEdit validate its high fidelity and editability in video editing while enhancing visual appeal and effectiveness.

## 2 RELATED WORKS

### 2.1 DIFFUSION-BASED GENERATIVE MODELS

Score-based generative frameworks (Vincent, 2011; Song et al., 2020b), particularly deep diffusion models (Ho et al., 2020; Song et al., 2020a), have shown promise in surpassing the previous state-of-the-art quality of generative adversarial networks (GANs) (Goodfellow et al., 2020). Diffusion models gradually denoise data and can restore the original or conditionally generate data based on input conditions. Especially, remarkable progress was made in image generation, where diffusion-based text-to-image (T2I) models (Ramesh et al., 2022; Saharia et al., 2022a) generate high-quality images from input text. The T2I models are currently extending their visual generative abilities into video to perform text-to-video (T2V) generation. Early work (Ho et al., 2022b) in T2V generation considers adding another axis on the T2I model to accommodate video data, and further works (Wu et al., 2022a; Hong et al., 2022) utilized pre-trained T2I model with large-scale text-to-image datasets (Deng et al., 2009). To ensure temporal consistency in video frames, temporal attention methods (Ho et al., 2022a; Singer et al., 2022) are also proposed. Many lines of diffusion-based models extend synthesis to various visual reconstructions including inpainting and super-resolution (Saharia et al., 2022b; Lugmayr et al., 2022). Among them, visual editing is facing new challenges in incorporating multi-modality and video domains, which will be discussed in more detail below.

### 2.2 IMAGE AND VIDEO EDITING

GAN has been a popular basis for image editing, and currently, the diffusion models have revolutionized it (Meng et al., 2021; Avrahami et al., 2022), offering a new framework for synthesis. Text-based image editing models aim to edit images based on free-text descriptions. To perform this, Prompt-to-Prompt (Hertz et al., 2022) proposes to train a model to generate an input image based on one prompt and to modify it conditioned on another prompt. To hold the fidelity of an input image, personalized fine-tuning methods (Valevski et al., 2022) are also introduced. Instruct-Pix2Pix (Brooks et al., 2022) incorporates pre-trained diffusion models and performs quick edits without fine-tuning. Along with the growth of image editing, there have been several efforts to perform video editing. Text2Live (Bar-Tal et al., 2022) presents zero-shot text-based editing for an image and videos. Although many works including Tune-A-Video (Wu et al., 2022b) Gen-1 (Esser et al., 2023), VideoComposer (Wang et al., 2023) and Control-A-Video (Chen et al., 2023) properly edit styles or objects in a video, they are limited to rigid types of editing (*i.e.*, style transfer, object overlay) and still challenging to dynamic motion change or seamless temporal editing. Thus, PDEdit first performs spatial-temporal non-rigid edits to a video based on the texts.

## 3 PRELIMINARIES

### 3.1 DENOISING DIFFUSION PROBABILISTIC MODELS

Denoising diffusion probabilistic models (DDPMs) (Ho et al., 2020) are probabilistic generative models trained to reconstruct a sequence of data $x_1, \cdots, x_T$ satisfying Markov chain. For a given data distribution $x_0 \sim q(x_0)$, the Markov transition is defined as given below, assuming Gaussian distribution and pre-defined variance $\beta_t$ of scheduling diffusion process:

$$q(x_t|x_{t-1}) = \mathcal{N}(x_t; \sqrt{1-\beta_t}x_{t-1}, \beta_t\mathbb{I}) \qquad t = 1, \cdots, T, \tag{1}$$

where this process is adding Gaussian noise gradually up to the distribution of $x_T$, which is referred to as a *forward process* of the diffusion process. The reason that we first perform the forward process is the distributions in this process can be used as information for training our DDPMs in the *reverse process*. Thanks to the Bayes' rule and Markov property, [2] we further derive an intuitive format of the forward process as conditional probabilities with the input of original data $x_0$ as given below:

$$q(x_t|x_0) = \mathcal{N}(x_t; \sqrt{\bar{\alpha}_t}x_0, (1-\bar{\alpha}_t)\mathbb{I}),$$

$$q(x_{t-1}|x_t, x_0) = \mathcal{N}(x_{t-1}; \tilde{\mu}_t(x_t, x_0), \tilde{\beta}_t\mathbb{I}), \quad \tilde{\mu}_t(x_t, x_0) = \frac{\sqrt{\bar{\alpha}_{t-1}}\beta_t}{1-\bar{\alpha}_t}x_0 + \frac{\sqrt{\alpha_t}(1-\bar{\alpha}_{t-1})}{1-\bar{\alpha}_t}x_t \tag{2}$$

where $\alpha_t = 1 - \beta_t$, $\bar{\alpha}_t = \Pi_{s=1}^t \alpha_s$, and $\tilde{\beta}_t = \frac{1-\bar{\alpha}_{t-1}}{1-\bar{\alpha}_t}\beta_t$. In the reverse process the DDPMs generate the Markov chain of data $x_1, \cdots, x_T$ from prior distribution $p(x_T) = \mathcal{N}(x_T; 0, \mathbb{I})$ by utilizing Gaussian transitions as $p_\theta(x_{t-1}|x_t) = \mathcal{N}(x_{t-1}; \mu_\theta(x_t, t), \sigma_\theta(x_t, t))$, which finally aims to maximize the log-likelihood $log(p_\theta(x_0))$. To this, one can follow the variational inference, and detour this by maximizing the variational lower bound of the negative log-likelihood. This makes a closed-form of KL divergence among the distributions $p_\theta(x_{t-1}|x_t)$ and $q(x_{t-1}|x_t, x_0)$, while getting close together by optimizing the learnable parameter $\theta$. The beauty of DDPMs is that the whole process is summarized that the model is interpreted as a sequence of denoising autoencoders $\epsilon_\theta(x_t, t)$ and is trained to predict a denoising variant of input $x_t$ as given below:

$$\mathbb{E}_{x,\epsilon\sim\mathcal{N}(0,1),t\sim\mathcal{U}\{1,T\}}[||\epsilon - \epsilon_\theta(x_t, t)||_2^2]. \tag{3}$$

To hold the robustness in all steps of denoising encoders, the step $t$ is sampled from a discrete uniform distribution $\mathcal{U}\{1, T\}$ ranging up to maximum denoising step $T$.

### 3.2 TEXT-GUIDED DIFFUSION MODEL AND LATENT MODEL

Text-guided diffusion model is one of the conditional diffusion frameworks, where it aims to recover an output data $x_0$ from the $t$ step random noise $x_t$ under the condition of input text prompt $\mathcal{P}$. Following the DDPM, the network is also performed for the process of denoising encoder with the objective given as $\mathbb{E}_{z,\epsilon\sim\mathcal{N}(0,1),t\sim\mathcal{U}\{1,T\}}[||\epsilon - \epsilon_\theta(z_t, t, \mathcal{C})||_2^2]$. To give a deep multi-modal understanding between text and the data, a latent encoding $z_t = E(x_t)$ is introduced from pre-trained encoders (*e.g.*, VQ-VAE (Van Den Oord et al., 2017)) and also from large-scale pre-trained textual embedding $\mathcal{C} = \psi(\mathcal{P})$ (*e.g.*, CLIP (Radford et al., 2021)). To be specific, as we work on video data, the $z_t \in \mathbb{R}^{l \times p \times d}$ is $d$ dimensional features from video with a frame length $l$ and $p$ is the number of image patches in a single frame for patch-wise image feature quantization. The $\epsilon_\theta$ is the latent neural networks for video diffusion models (Ho et al., 2022b; Wu et al., 2022b; Singer et al., 2022) that have the capacity to consider temporal consistency among the frames.

### 3.3 DDIM SAMPLING AND INVERSION

To accelerate the reverse process of DDPM, denoising diffusion implicit model (DDIM) (Song et al., 2020a) is presented. DDIM samples latent features with a small number of denoising steps as below:

$$z_{t-1} = \sqrt{\frac{\alpha_{t-1}}{\alpha_t}}z_t + \left(\sqrt{\frac{1}{\alpha_{t-1}} - 1} - \sqrt{\frac{1}{\alpha_t} - 1}\right) \cdot \epsilon_\theta(z_t, t, \mathcal{C}). \tag{4}$$

DDIM sampling can also be reversed to make latent noise, which gives corresponding latent features as $z_{t+1} = \sqrt{\frac{\alpha_{t+1}}{\alpha_t}}z_t + \left(\sqrt{\frac{1}{\alpha_{t+1}} - 1} - \sqrt{\frac{1}{\alpha_t} - 1}\right) \cdot \epsilon_\theta(z_t, t, \mathcal{C})$ denoted as DDIM inversion process.

---

[2]See the detailed proof of reverse process in Appendix C.

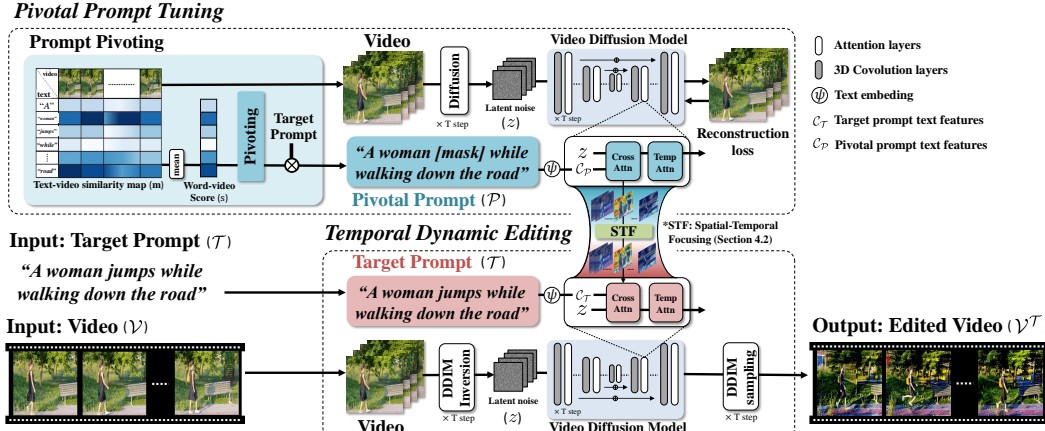

Figure 3: Pivotal Dynamic Editing (PDEdit) is composed of two processes: (1) pivotal prompt tuning that tunes an input video with pivotal prompt and (2) temporal dynamic editing that edits the input video conforming to the target prompt by applying spatial-temporal focusing.

# 4 PIVOTAL DYNAMIC EDITING

Text-based video editing aims to modify an input video in a way that accurately conveys the meaning of a given target text. To this, the video editing framework is typically built on a pre-trained text-to-video (T2V) model, which generates the necessary attributes for editing before they are incorporated into the desired video. In the formal definition of text-based video editing, the system takes inputs of video $\mathcal{V}$ and a target text prompt $\mathcal{T}$ and produces a modified video $\mathcal{V}^{\mathcal{T}}$ containing the intended textual meaning. In procedures, video editing systems involve two key processes: (1) tuning an input video and (2) editing the video. During the tuning process, the system is trained (*i.e.*, tuned) to generate the input video and understand the contextual meaning of the video. In the editing process, the system generates a modified video based on the input video by incorporating the attributes required in the target prompt, which shows the resulting video properly combining the given video and the attributes obtained from the knowledge in the pre-trained model. Under this process, in Figure 3, we propose the pivotal dynamic editing (PDEdit) framework, which contains two main processes of tuning and editing referred to as (1) Pivotal Prompt Tuning and (2) Temporal Dynamic Editing. The details of each process are provided in the following.

## 4.1 PIVOTAL PROMPT TUNING

Objectives for the video tuning process are two folds: (1) understanding the contextual meaning in the input video and (2) controlling the model to generate input video with high fidelity. Current systems (Hertz et al., 2022; Wu et al., 2022b) have introduced additional text prompt (*i.e.*, source prompt) that describes the input video and utilize it to control the model to generate the input video according to the source prompt. Although these prompts are primarily focused on understanding the video, there were not much of concerns about which prompts are more effective in controlling the model. Furthermore, the resulting edits were vulnerable to the variants of the source prompts. To this end, our solution is to deduce a prompt for tuning based on the target prompt. We refer to the process of deducing a prompt from a target prompt as *prompt pivoting*, where it aims to modify a target prompt into another prompt that has more alignment with the input video while preserving textual resemblance. Figure 3 presents a pivotal prompt $\mathcal{P}$ that we obtained by applying prompt pivoting to the target prompt. The pivotal prompt is reasonable as (1) the system no longer relies on the source prompt and (2) it provides more effective conditions to control the model. By ensuring a resemblance to the target prompt, the editing model can generate a video with high fidelity to the input video while remaining the key differences required for effective editing. In the following, our technical contribution is to introduce two prompt pivoting methods formulated as $\mathcal{P} = f(\mathcal{T}, \mathcal{V})$ (*i.e.*, $f$: prompt pivoting): (1) Editing Factor Pivoting and (2) Distributional Pivoting.

**Editing Factor Pivoting.** As a target prompt describes the context to be changed from the input video, the prompt is composed mainly of two groups of words: (1) words aligned with the input video and (2) words unrelated to the input video that contribute to editing. Intrigued by this, we define the words that contribute to editing as 'editing factor' denoting $\mathbf{w}_k = \{w_1, \cdots, w_k\}$, where the $k$ is the number of words in the editing factor. Thus, our proposed prompt pivoting aims to identify the $\mathbf{w}_k$ and build a pivotal prompt by disentangling them from the target prompt. In detail, to identify the editing factor, we introduce the similarity-based selection by calculating word-video clip (Radford et al., 2021) scores. We first embed the video and target prompt into $d$-dimensional features as $\mathbf{v} = \psi_v(\mathcal{V}) \in \mathbb{R}^{l \times d}$ and $\mathbf{t} = \psi_t(\mathcal{T}) \in \mathbb{R}^{m \times d}$, where the $\psi_v(\cdot), \psi_t(\cdot)$ are clip embedding for image and word, and $l, m$ are the numbers of frames and words. As shown in Figure 3, we build a clip similarity score map as $\mathbf{m} = \mathbf{t}\mathbf{v}^\top \in \mathbb{R}^{m \times l}$ and take an average pooling along the frame axis and obtain word-video scores as $\mathbf{s} = \{s_1, \cdots, s_m\}$. The words with low similarities denote that they are not related to the input video and are rather close to words for editing, thus we regard them as editing factors as $\mathbf{w}_k = w_{\mathrm{argmin}(\mathbf{s},k)}$. The $\mathrm{argmin}(\mathbf{s}, k)$ is the indices of bottom-$k$ words in terms of the word-video score $\mathbf{s}$. To disentangle the $\mathbf{w}_k$ from the target prompt $\mathcal{T}$, we apply a pad token mask on $\mathbf{w}_k$ in $\mathcal{T}$ and finally build a pivotal prompt $\mathcal{P}$. This process can be formally summarized as $\mathcal{P} = f_k(\mathcal{T}, \mathcal{V})$, where $f_k$ is the editing factor pivoting and the $k$ denotes bottom-$k$ words to mask.

**Distributional Pivoting.** Although the editing factor pivoting is quite intuitive, there exists a heuristic that the number of editing factors is fixed to the number $k$. To avoid the heuristic, we introduce text-video similarity score distributions $\mathcal{D} \supset \{\mathcal{X}, \mathcal{Y}\}$ composed of positive $\mathcal{X}$ and negative $\mathcal{Y}$ distributions in Figure 4. Here, the positive contains the scores of the text-video pairs describing each other and the negative contains the scores when texts are not related to videos. Based on two distributions, we define a deterministic score $s^* \in \mathbb{R}$ to identify the editing factor. In detail, our study sets $s^*$ at the score where $\mathcal{X}$ and $\mathcal{Y}$ intersect[3] in Figure 4 and the words in target prompt are determined as editing factor if their word-video scores $\mathbf{s}$ are lower than the score $s^*$. We formulate it as distributional editing factor $\mathbf{w} = w_{\mathrm{arg}(\mathbf{s}<s^*)}$, where the $\mathrm{arg}(\mathbf{s} < s^*)$ is the indices of the scores lower than $s^*$. We summarize the whole process as

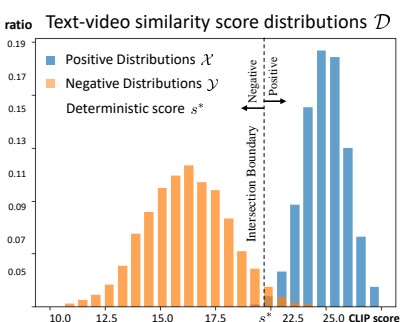

Figure 4: Clip score distributions of positive and negative text-video pairs.

$\mathcal{P} = f_{s^* \sim \mathcal{D}}(\mathcal{T}, \mathcal{V}, s^*)$, where $f_{s^* \sim \mathcal{D}}$ is a distributional pivoting process with a pad mask. To collect the $\mathcal{D}$, we use Charades-STA (Gao et al., 2017) dataset that contains videos and their descriptions of various scenes (*i.e.*, See examples of text-video pairs in Appendix D). The $\mathcal{X}$ is obtained from texts and their annotated video moments and $\mathcal{Y}$ is from texts and the other videos in the dataset.

**Video tuning.** Based on the pivotal prompt $\mathcal{P}$, we tune an input video $\mathcal{V}$ into our text-to-video (T2V) model. We first prepare noise features $z$ of the input video using diffusion, and the model is trained to reconstruct the original video by denoising the $z$, where the pivotal prompt $\mathcal{P}$ is conditioned in the denoising process. This denotes that our T2V model is tuned to generate original video with the input of the pivotal prompt. For the details of the T2V model, similar to work (Ho et al., 2022b), we extend 2D pre-trained diffusion model (*i.e.*, Stable Diffusion) into 3D with a temporal domain by modifying 2D convolution and attention layers. We inflate $3 \times 3$ kernels in 2D convolution layers to $1 \times 3 \times 3$ kernels as pseudo-3D convolution layers and append temporal attention in each attention layer to give temporal consistency. The text conditioning is performed on cross attention between noise features $z$ and pivotal prompt features $\mathcal{C}_\mathcal{P}$. As explained in Section 3.2, the model is trained to reconstruct video under the objective $\mathbb{E}_{z, \epsilon \sim \mathcal{N}(0,1), t \sim \mathcal{U}\{1, T\}}[||\epsilon - \epsilon_\theta(z_t, t, \mathcal{C}_\mathcal{P})||_2^2]$, where $\mathcal{C}_\mathcal{P} = \psi(\mathcal{P})$ is the pivotal prompt embedding and $z_t$ is $t$-step noise feature in denoising process.

## 4.2 TEMPORAL DYNAMIC EDITING

The video editing works in a way that the tuned T2V model infers a video based on input video when a conditional text is the target prompt. Thus, it is crucial to recognize the difference between the

---

[3]Score $s^*$ is such that $|\tilde{f}_\mathcal{X}(s^*) - \tilde{f}_\mathcal{Y}(s^*)| < \delta$, where $\tilde{f}_\mathcal{X}, \tilde{f}_\mathcal{Y}$ is approximation of the true probability density functions $f_\mathcal{X}, f_\mathcal{Y}$ by binning. $\delta = 0.01$ is chosen when $s^*$ approximates the unique intersection, $f_\mathcal{X}(s) = f_\mathcal{Y}(s)$

prompt (*i.e.*, $\mathcal{P}$) used for tuning and the target prompt $\mathcal{T}$, and then the model reflects the difference in generating the video. Our pivotal prompt contributes to a sensible edit because the resemblance to the target prompt enables the model to generate a video with high-fidelity to input video while preserving the key differences (*i.e.*, editing factor) needed for effective editing. However, our studies found that non-rigid editing (*e.g.*, motion change) is still challenging and unsatisfactory. Therefore, we devise temporal dynamic editing that modulates the influence of the editing factor on our T2V model. We break this down into two functionalities: (1) Spatial Focusing and (2) Temporal Focusing.

**Spatial-Temporal Focusing.** Spatial focusing aims to enhance the editing effect in the spatial domain. As shown in Figure 5, we build spatial focusing by reweighting the attention of editing factor $\mathbf{w}$ in cross attention map between the target prompt $\mathcal{T}$ and the latent noise $z$. Thus, we first build the cross attention map as $\mathbf{x}^i = C_{\mathcal{T}}(z^i)^{\top}/\sqrt{d} \in \mathbb{R}^{m \times p}$ by taking $C_{\mathcal{T}}, z^i$ as query and key of attention (Vaswani et al., 2017), where $C_{\mathcal{T}} = \psi(\mathcal{T}) \in \mathbb{R}^{m \times d}$ is target prompt features, $z^i \in \mathbb{R}^{p \times d}$ is $i$-th frame latent feature and $p$ (*e.g.*, 64 x 64) is the number of patches for image quantization. Our reweighting incorporates the following two weightings: (1) editing weighting and (2) fidelity weighting. The fidelity weighting utilizes attention map of the pivotal prompt $\mathcal{P}$ as $\mathbf{y}^i = C_{\mathcal{P}}(z^i)^{\top}/\sqrt{d} \in \mathbb{R}^{m \times p}$, which allows controlling the fidelity in editing process. Therefore the spatial focusing is formulated as $\mathbf{x}^i = \alpha_{\mathbf{w}} \odot \mathbf{x}^i + \beta_{\mathbf{w}} \odot \mathbf{y}^i$, where

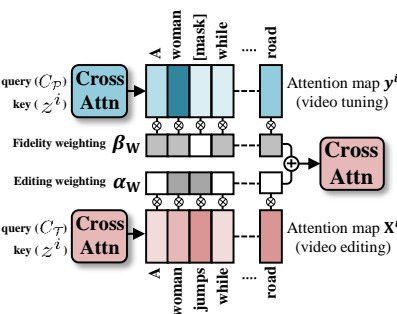

Figure 5: Conceptual illustration of spatial focusing process.

$\alpha_{\mathbf{w}} = \{0, 0, \alpha, \cdots\} \in \mathbb{R}^m$ denotes editing coefficient ($1 < \alpha < 2$) corresponding to the position of editing factor $\mathbf{w}$, $\beta_{\mathbf{w}} = \{\beta, \beta, 0, \cdots\} \in \mathbb{R}^m$ is fidelity coefficient ($0 < \beta < 1$) and $\odot$ is element-wise multiplication with broadcasting. Especially, when the editing factor includes a 'predicate', we consider the target prompt requires motion change and assign further weights in $\alpha_{\mathbf{w}}$ to the words about the subject[4] to focus on the subject of the motion. Certain motion edits (*e.g.*, jump, turn) require temporary changes. To address this, we introduce temporal weights $\gamma \in \mathbb{R}^l$ with frame length $l$ as $\mathbf{x}^i = \gamma^i(\alpha_{\mathbf{w}} \odot \mathbf{x}^i) + (1 - \gamma^i)(\beta_{\mathbf{w}} \odot \mathbf{y}^i)$, which smoothly blend editing along the frames. Figure 6 shows an example of temporal focusing using a Gaussian curve.

**Video editing.** Founded on spatial-temporal focusing, PDEdit performs denoising on initial latent noise obtained from input video using DDIM inversion. For denoising, our diffusion-based T2V model predicts the added noise. Using the predicted noise, DDIM sampling performs step-wise denoising as given in Equation 4, which is formulated as $\mathcal{V}^{\mathcal{T}} = \text{T2V}(\text{DDIM}_{\text{inv}}(\mathcal{V}), \mathcal{T})$.

## 5 Applications of Pivotal Dynamic Editing Framework

In this section, we showcase various editing results on our proposed PDEdit framework.

**Object overlay and Style transfer.** The first and the second videos of Figure 6 are about the video according to the rigid edits. Utilizing the knowledge of the pre-trained T2I model and PDEdit properly changes targets (*i.e.*, object or background) in the video while maintaining frame consistency under our proposed temporal frame attention. Especially in the video of the last row, the background of a woman walking is changed to an environment similar to an airport, while preserving the woman's walking motion. Further applications of our PDEdit are also available in Appendix F.

**Motion change and Temporal change.** The video in the fourth row of Figure 6 shows the editing results from the input video according to the non-rigid edits. The non-rigid edits such as dynamic motion change are more challenging than rigid-type free editing (*e.g.*, style transfer, object overlay) as they require predicting and generating movements within a given range of possibilities. PDEdit changes motion in a way that a walking woman dances at the same time, conforming to the target text "A woman dances ballet while walking on the road." In the last row's video, PDEdit demonstrates

---

[4]Motion change is usually performed by subjective in the target text prompt. Appendix E provides more details about finding the subjective in a text.

**Input video**

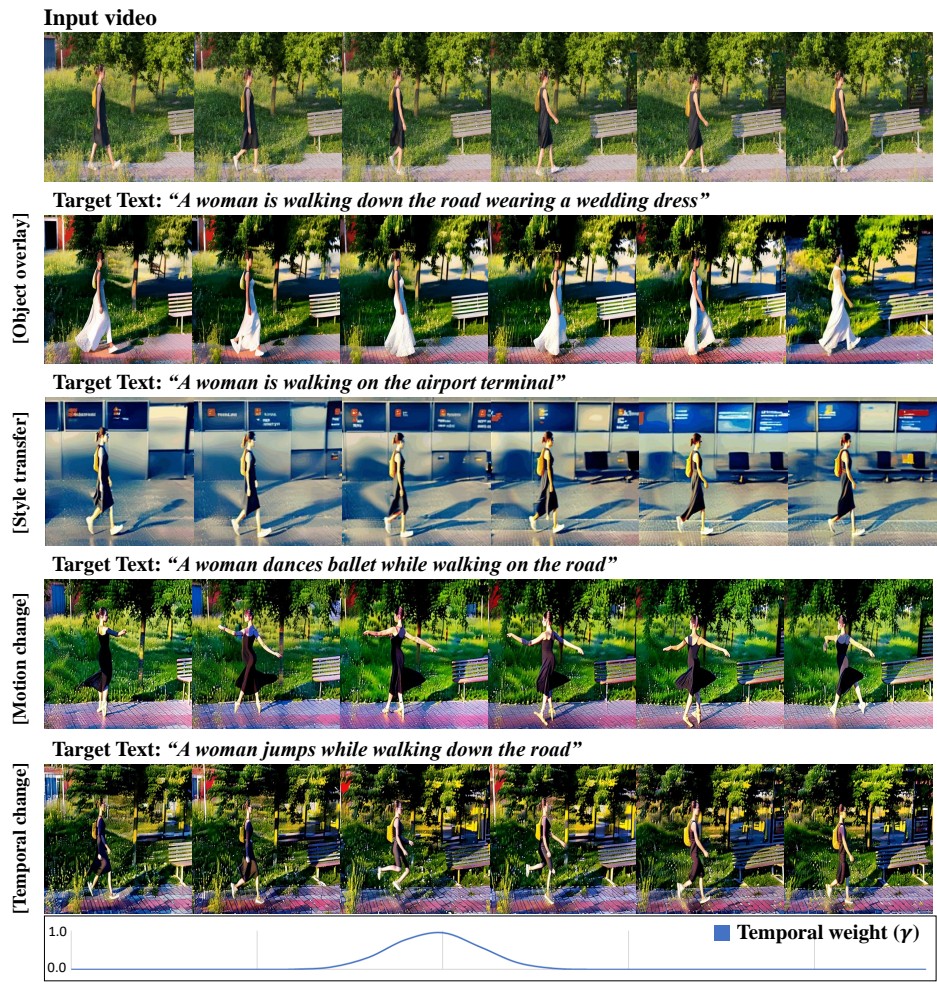

Figure 6: Sample results of applications with PDEdit in terms of different edits (object overlay, style transfer, motion change, temporal change). Best viewed in zoom.

its ability to edit temporal motion. When the target prompt describes "jumping" while a woman is walking, temporal changes are necessary to depict the jumping action realistically. Our temporal focusing approach is used to design the natural jumping motion in this scenario.

## 6 EXPERIMENTS

### 6.1 BASELINE COMPARISONS

**Dataset and Baseline.** We select 21 videos from DAVIS (Pont-Tuset et al., 2017), which contain a variety of objects and scenes spanning about 50 to 70 frames. We introduce target prompts that require both rigid and non-rigid edits on the input videos. We compare baselines on their public codes[5]: Text2Video-Zero (T2V-Zero) (Khachatryan et al., 2023) zero-shot video editing to given target text, Tune-A-Video (TAV) (Wu et al., 2022b) recent text-based video generation to edit video.

**Qualitative and Quantitative Results.** Figure 7 (a) and (b) show visual comparisons of two typical edits: (1) motion change and (2) object overlay. PDEdit accurately represents the rolling down action in motion change. In object overlay, all three models make edits, but PDEdit maintains the highest fidelity to the input video, as indicated by the red box. We conducted quantitative evaluations

---

[5]https://github.com/Picsart-AI-Research/Text2Video-Zero; https://github.com/showlab/Tune-A-Video

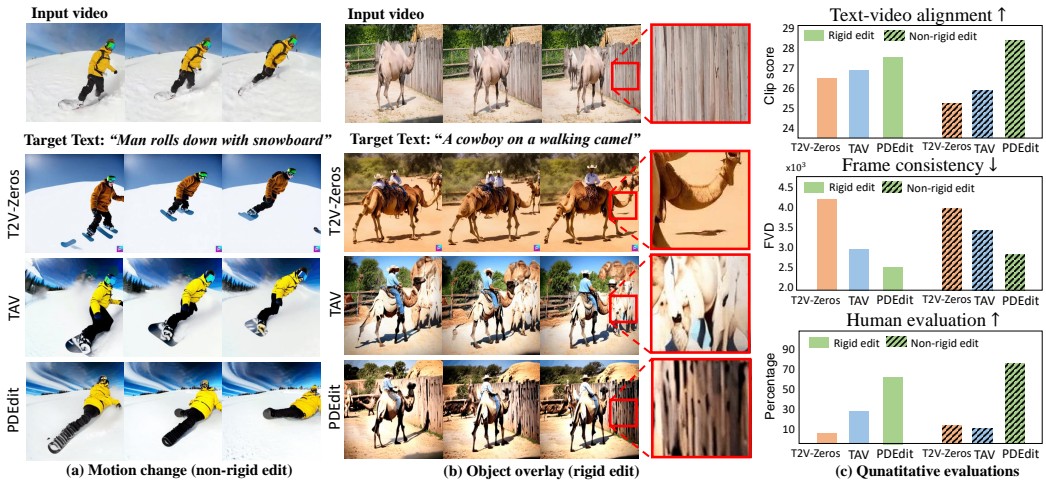

Figure 7: Qualitative and quantitative comparisons to previous text-based video editing systems. Further results on more metrics with different types of datasets are also available in Appendix F.

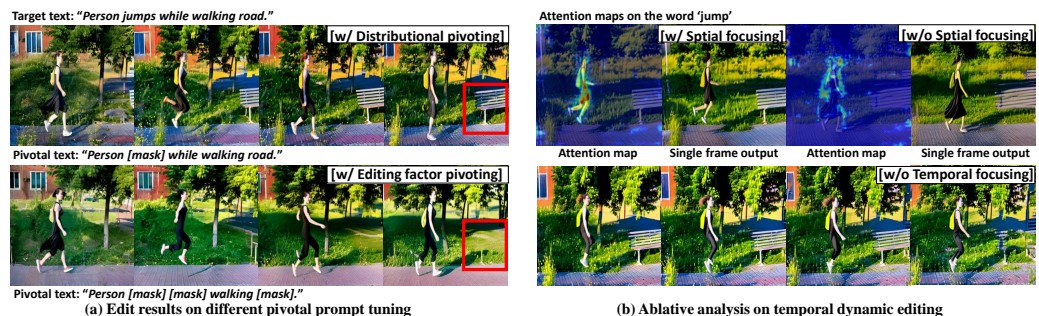

Figure 8: Ablation studies of pivotal prompting and temporal dynamic editing on input in Figure 6.

comparing our edits to previous baselines. In Figure 7(c), we used clip similarity scores to assess text-video alignment. PDEdit outperforms others, especially in non-rigid editing. We also measure Fréchet Video Distance (FVD) (Unterthiner et al., 2019) for frame consistency, where both PDEdit and TAV excelled in non-rigid edits, but TAV's edits do not align well with the target text. In human evaluation, with 36 participants rating the edits for preference, PDEdit received higher scores.

## 6.2 ABLATION STUDIES

Ablation studies were conducted on two prompt pivoting methods (editing factor and distributional) for pivotal prompt tuning, and two focusing methods (spatial and temporal) for temporal dynamic editing. In Figure 8 (a), both pivoting methods perform editing, but the distributional appears more effective for reconstruction. The editing factor sometimes mistakes due to the fixed number of words (*e.g.*, k=3), affecting unsatisfied reconstruction (*e.g.*, "bench" in the red box). Focusing modules enhance editing effectiveness. In Figure 8 (b), spatial focusing enhances dynamic editing by highlighting the object in the attention map related to the editing factor (*e.g.*, "jump"). Without temporal focusing, editing related to temporal motion changes, as seen with levitation, is not successful.

## 7 CONCLUSION

Pivotal Dynamic Editing (PDEdit) framework for text-based video editing that allows spatial-temporal non-rigid edits to a general video using a single target text. It can change the motion of objects in a video at specific moments or throughout the entire video while maintaining temporal consistency. PDEdit is versatile and effective for editing various types of videos.

ETHICS STATEMENT

PDEdit proposes general framework for text-based video editing, where we showcase several possible applications of our framework. Although editing is designed to generate an appropriate image or video corresponding to the user's needs, but the highly advanced editing technology can result in several societal negative impacts such as fake material for commercial profit and privacy issues. Therefore it is also required to build regulations (*e.g.*, Learning-based forensic analysis, Digital watermarking) against indiscriminate abuse in editing or generative models in the future.

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

## A  IMPLEMENTATION DETAILS

PDEdit framework is based on pre-trained image diffusion model Rombach et al. (2022) with public pre-trained weights[6] The input video is uniformly sampled by 24 frames with a resolution of 512 x 512. For tuning a video, it takes about 200 steps with a single sample (*i.e.*, batch size of 1) and a learning rate of 3 x 10-5, which corresponds to 8 minutes of our NVIDIA A100 GPU. For editing a video, we utilize DDIM sampler Song et al. (2020a) with classifier-free guidance Ho & Salimans (2022), which takes 2 minute for operation. The text feature embedding utilizes the pre-trained CLIP model (ViT-L/14) Radford et al. (2021) and the image feature embedding utilizes Variational Auto Encoder Kingma & Welling (2013) for latent feature representation.

## B  ANALYSIS ON VIDEO EDITING RESULTS OF CURRENT EDITING SYSTEMS

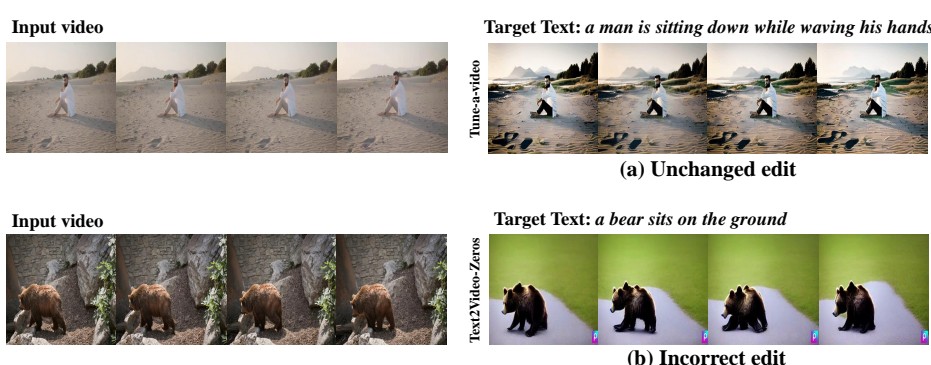

Figure 9: Failure cases of current video editing systems

Figure 9 shows that the current editing systems Wu et al. (2022b); Khachatryan et al. (2023) are not available to perform non-rigid edits (*i.e.*, motion change, temporal change) by resulting in unchanged edits or incorrect edits.

## C  PROOF FOR REVERSE DIFFUSION PROCESS

For the proof of $q(x_{t_1}|x_t, x_0)$, we apply Bayes' rule, as given below:

$$q(x_{t-1}|x_t, x_0) = q(x_t|x_{t-1}, x_0)\frac{q(x_{t-1}|x_0)}{q(x_t|x_0)},$$

$$\propto \exp(-\frac{1}{2}(\frac{(x_t - \sqrt{\alpha_t}x_{t-1})^2}{\beta_t}) + \frac{(x_{t-1} - \sqrt{\bar{\alpha}_{t-1}}x_0)^2}{1 - \bar{\alpha}_{t-1}} - \frac{(x_t - \sqrt{\bar{\alpha}}x_0)^2}{1 - \bar{\alpha}_t}))$$

$$=\exp(-\frac{1}{2}(\frac{x_t^2 - 2\sqrt{\alpha_t}x_t x_{t-1} + \alpha_t x_{t-1}^2}{\beta_t} + \frac{x_{t-1}^2 - 2\sqrt{\bar{\alpha}_{t-1}}x_0 x_{t-1} + \bar{\alpha}_{t-1}x_0^2}{1 - \bar{\alpha}_{t-1}} - \frac{(x_t - \sqrt{\bar{\alpha}_t}x_0)^2}{1 - \bar{\alpha}_t})),$$

$$=\exp(-\frac{1}{2}((\frac{\alpha_t}{\beta_t} + \frac{1}{1 - \bar{\alpha}_{t-1}})x_{t-1}^2 - (\frac{2\sqrt{a_t}}{\beta}x_t + \frac{2\sqrt{\bar{\alpha}_{t-1}}}{1 - \bar{\alpha}_{t-1}}x_0)x_{t-1} + C(x_t, x_0))),$$

$$(5)$$

where the $C(x_t, x_0)$ can be interpreted as function not involving $x_{t-1}$. For the $\tilde{\mu}_t(x_t, x_0)$, following the Gaussian density function, we have the equations in terms of mean and variance parameterized

---

[6]https://huggingface.co/runwayml/stable-diffusion-v1-5

as given below:

$$\tilde{\beta}_t = (\frac{\alpha_t}{\beta_t} + \frac{1}{1 - \bar{\alpha}_{t-1}})^{-1} = (\frac{\alpha_t - \bar{\alpha}_t + \beta_t}{\beta_t(1 - \bar{\alpha}_{t-1})})^{-1} = \frac{1 - \bar{\alpha}_{t-1}}{1 - \bar{\alpha}_t}\beta_t$$

$$\tilde{\mu}_t(x_t, x_0) = (\frac{\sqrt{\alpha_t}}{\beta_t}x_t + \frac{\sqrt{\bar{\alpha}_{t-1}}}{1 - \bar{\alpha}_{t-1}}x_0)/(\frac{\alpha_t}{\beta_t} + \frac{1}{1 - \bar{\alpha}_{t-1}})$$

$$= (\frac{\sqrt{\alpha_t}}{\beta_t}x_t + \frac{\sqrt{\bar{\alpha}_{t-1}}}{1 - \bar{\alpha}_{t-1}}x_0)\frac{1 - \bar{\alpha}_{t-1}}{1 - \bar{\alpha}_t}\beta_t \qquad (6)$$

$$= \frac{\sqrt{\bar{\alpha}_{t-1}}\beta_t}{1 - \bar{\alpha}_t}x_0 + \frac{\sqrt{\alpha_t}(1 - \bar{\alpha}_{t-1})}{1 - \bar{\alpha}_t}x_t$$

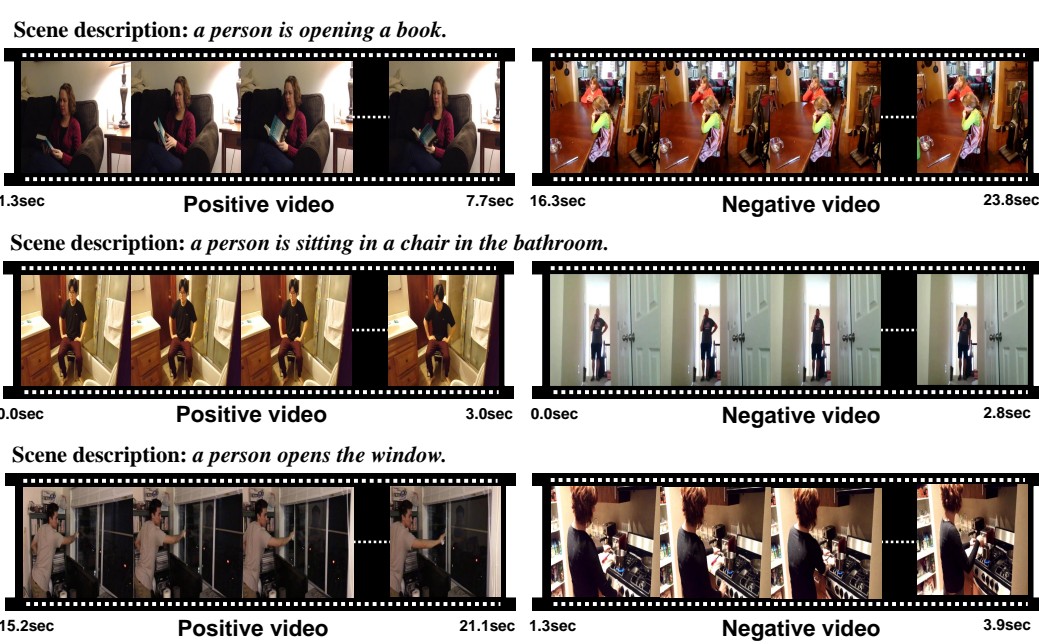

Figure 10: Samples in Charades-STA dataset, where we build the positive text-video pairs and negative pairs for constructing positive score distributions and negative distributions.

## D SPECIFICATIONS AND EXAMPLES OF CHARADES-STA DATASET

Charades-STA Gao et al. (2017) dataset includes about 30 seconds of videos for human behaviors and scene descriptions of language queries. The dataset contains 12,400 video-text pairs and 3700 pairs for testing and all the pairs have their temporal annotations (*i.e.*, start-time, end-time) in video corresponding to the text. Figure 10 provides the illustrations of videos in Charades-STA and we leverage the dataset to measure clip scores by building positive text-video pairs and negative pairs. The positive pairs are obtained from text and annotated moments in the video of each pair, while the negative pairs are obtained from the same text but other annotated moments in other videos in the dataset under random selection.

## E IDENTIFYING A SUBJECT IN TARGET PROMPT FOR EDITING MOTION

As an editing about motion change requires a modification in visual attributes pertaining to the subject of the motion, we provide a further weighting to the attention map corresponding to the subject. In detail, to investigate whether the target prompt requires editing about motion change, we scrutinize that the editing factors include "predicate" (*i.e.*, predicate usually contains information about motion). If the editing factor includes a predicate, the editing is regarded as motion change and further weights are given to the attention map of subjective in the target prompt. Here, we

utilize the part of speech (POS) tagger provided by the natural language toolkit Loper & Bird (2002) for identifying predicate and subjective in the target prompt. As a specific example, for the target text prompt (*i.e.*, "A woman jumps while walking down the road") provided in Figure 3 of the paper, the editing factor was selected as $\mathbf{w} = \{\text{"}jumps\text{"}\}$, thus the editing factor weighting is initially constructed as $\alpha_{\mathbf{w}} = \{0, 0, \alpha, 0, 0, 0, 0, 0\}$ and fidelity weighting is constructed as $\beta_{\mathbf{w}} = \{\beta, \beta, 0, \beta, \beta, \beta, \beta, \beta\}$. Henceforth, the editing is about motion change, thus we give further weight to the subjective (*i.e.*, "woman"), updating as $\alpha_{\mathbf{w}} = \{0, \alpha, \alpha, 0, 0, 0, 0, 0\}$.

# F ADDITIONAL RESULTS

| | Alignment | Consistency | | | Fidelity | |
|---|---|---|---|---|---|---|
| | CLIP$^\star$ ↑ | CLIP$^\dagger$ ↑ | FVD ↓ | PSNR ↑ | LPIPS ↓ | SSIM ↑ |
| PDEdit | 26.1 / 27.9 | 93.1 / 94.2 | 2831 / 2521 | 19.75 / 21.24 | 0.367 / 0.312 | 0.713 / 0.782 |
| *w.o.* PPT | 24.3 / 25.5 | 92.5 / 93.9 | 3161 / 2916 | 15.37 / 17.12 | 0.453 / 0.443 | 0.611 / 0.635 |
| *w.o.* STF | 25.9 / 27.2 | 90.8 / 92.2 | 3552 / 2864 | 18.86 / 20.13 | 0.375 / 0.346 | 0.656 / 0.684 |
| TAV | 16.4 / 26.0 | 89.8 / 92.6 | 3403 / 2720 | 14.62 / 17.46 | 0.577 / 0.445 | 0.542 / 0.627 |
| T2V-Zero | 13.7 / 24.9 | 86.1 / 87.4 | 4052 / 4235 | 9.31 / 11.73 | 0.590 / 0.573 | 0.409 / 0.426 |
| Video-P2P | 15.1 / 26.5 | 91.6 / 93.4 | 3261 / 2683 | 16.17 / 18.46 | 0.450 / 0.395 | 0.583 / 0.718 |
| Pix2Video | 15.9 / 25.8 | 90.4 / 91.8 | 3131 / 2704 | 16.09 / 18.31 | 0.496 / 0.421 | 0.561 / 0.729 |

Table 1: Quantitative evaluations about edited videos based on DAVIS dataset in terms of non-rigid/rigid type editing corresponding to textual alignment (Alignment), frame consistency (Consistency), and fidelity. PPT: Pivotal Prompt Tuning, STF: Spatial-Temporal Focusing, CLIP$^\star$: text-video clip score, CLIP$^\dagger$: image-image clip score, FVD: fréchet video distance, PSNR: peak signal-to-noise ratio, LPIPS: learned perceptual image patch similarity, SSIM: structural similarity index measure. Video-P2P (Liu et al., 2023) and Pix2Video (Ceylan et al., 2023) are reproduced by their public codes.

| | Alignment | Consistency | | | Fidelity | |
|---|---|---|---|---|---|---|
| | CLIP$^\star$ ↑ | CLIP$^\dagger$ ↑ | FVD ↓ | PSNR ↑ | LPIPS ↓ | SSIM ↑ |
| DAVIS | 27.4 | 93.7 | 2584 | 20.94 | 0.318 | 0.762 |
| UCF101 | 26.7 | 92.6 | 2399 | 18.63 | 0.258 | 0.596 |
| WebVid-10M | 26.4 | 92.8 | 2857 | 17.95 | 0.298 | 0.620 |

Table 2: Quantitative evaluations about resulting videos in various video datasets: DAVIS (Pont-Tuset et al., 2017), UCF101 (Soomro et al., 2012), WebVid-10M (Bain et al., 2021).

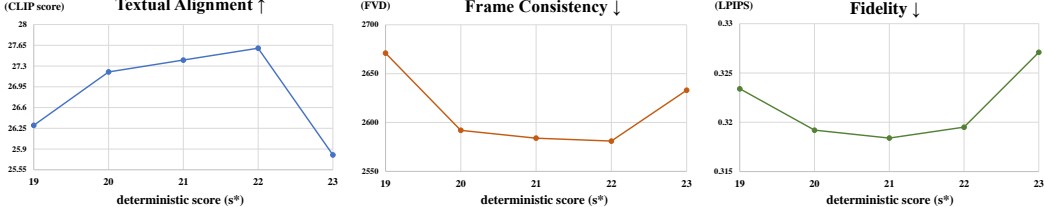

Figure 11: Sensitivity analysis of PDEdit according to deterministic score $s^*$.

**Quantitative Results** Table 1 provides a concise overview of our experimental results for edited videos, focusing on three key aspects: textual alignment, frame consistency, and fidelity. We measure textual alignment by computing the average CLIP score between the target prompt and the edited video frame. For frame consistency, we consider both the average CLIP score among sequential frames and the FVD between input and output videos. To assess fidelity to the input videos, we mask the edited regions in the edited videos and measure PSNR, LPIPS, and SSIM scores. Notably, across all these metrics, our editing system, PDEdit, demonstrates superior performance compared

to previous editing systems. Furthermore, in the realm of non-rigid editing, PDEdit exhibits performance comparable to traditional rigid-edit methods, even outperforming other models that were not successful in this context. In Table 2, To validate different types of video domains, we introduce 21 videos from three video datasets (DAVIS, UCF101, and WebVid-10M), where similar performances show that PDEdit operates generally without being domain dependent. In Figure 4, the discernible separation is evident between positive and negative pair about similarity score distributions. This distinction enables the establishment of a deterministic score, which effectively classifies the two distributions, thereby mitigating unstable editing factors. To investigate the stability of $s^*$, in Figure 11, we present editing performances according to deterministic scores $s^*$, where the stability region of $20 < s^* < 22$ is explicitly specified.

**Qualitative Results**  Videos used in additional results are from DAVIS Pont-Tuset et al. (2017) and materials with free copyright. (*i.e.*, https://www.youtube.com/watch?v=P4GB4t8sODU&t=456s, https://www.youtube.com/watch?v=9V52OxjXQ7A&t=484s)

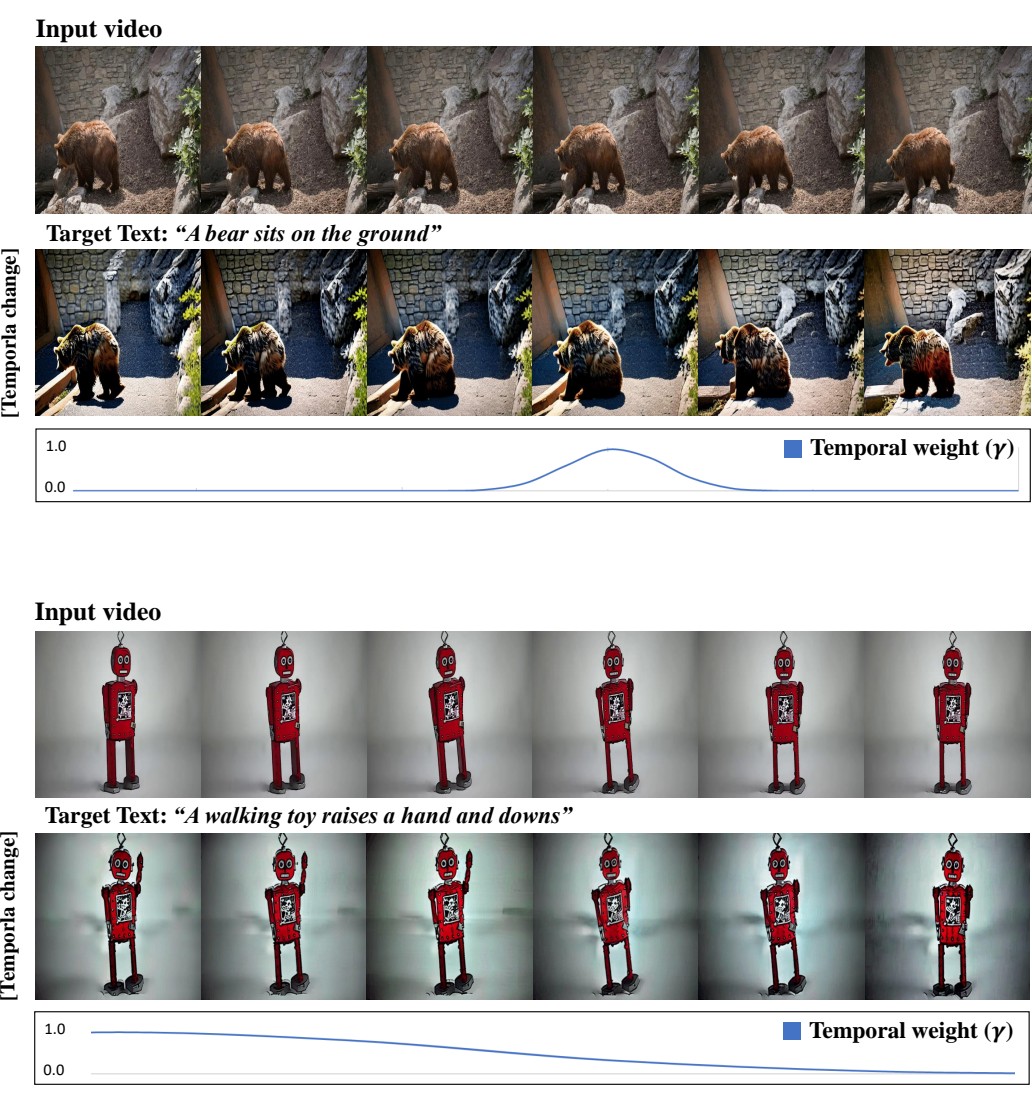

Figure 12: Edited results about temporal change.

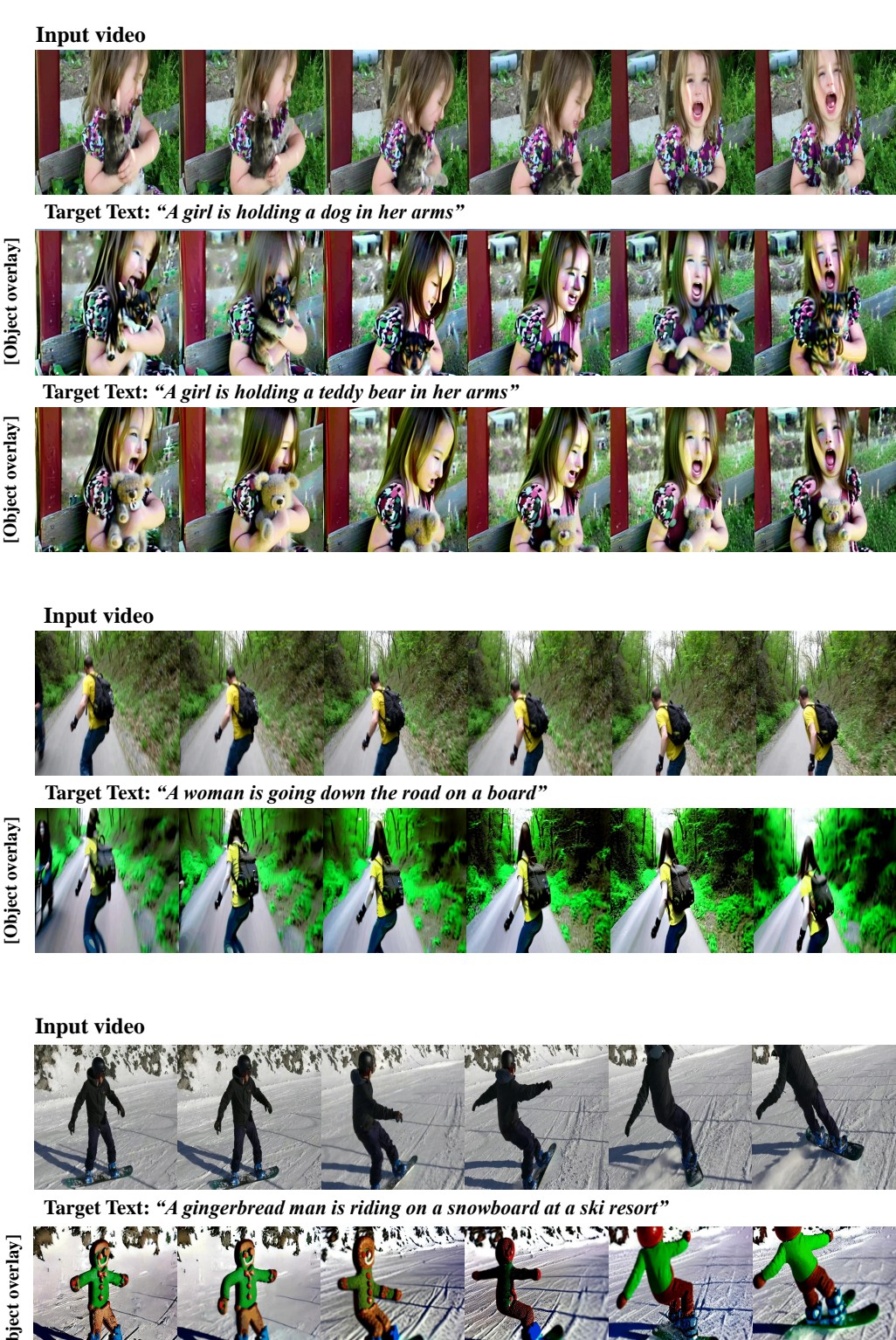

Figure 13: Edited results about object overlay.

**Input video**

**Target prompt:** *"An astronaut is jumping on the moon"*

[Motion change]

**Input video**

**Target Text:** *"A black swan is walking on the road"*

[Motion change]

**Input video**

**Target Text:** *"A man is praying"*

[Motion change]

Figure 14: Edited results about motion change.

**Input video**

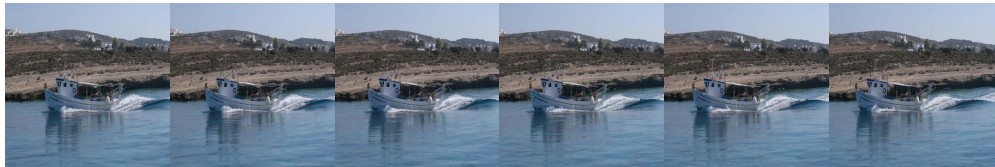

**Target Text:** *"A sailing boat is passing over the snowy sea"*

[Style transfer]

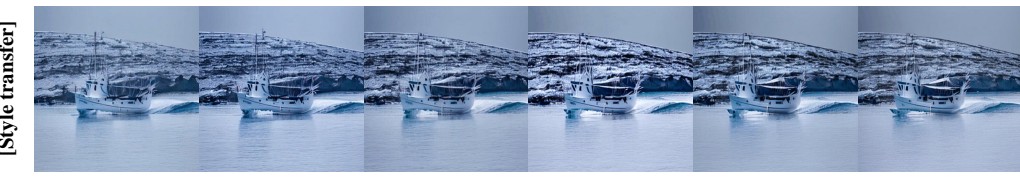

**Target Text:** *"A LEGO boat is passing over the river"*

[Style transfer]

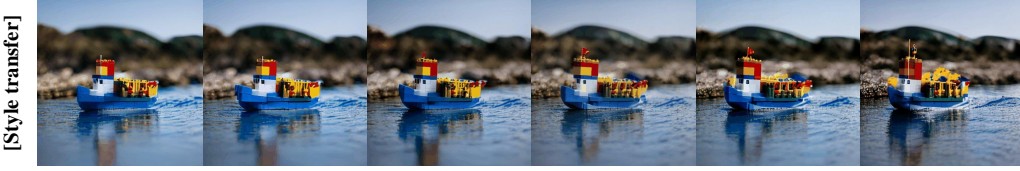

**Input video**

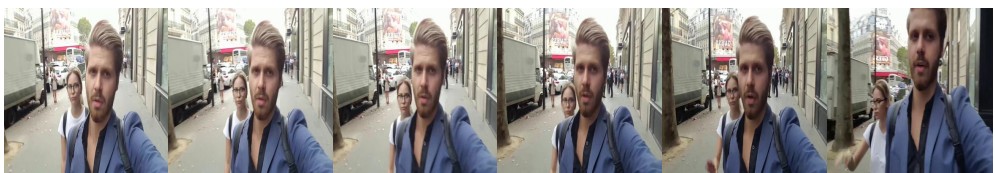

**Target Text:** *"Young boy and young girl take selfies while walking down the forest"*

[Style transfer]

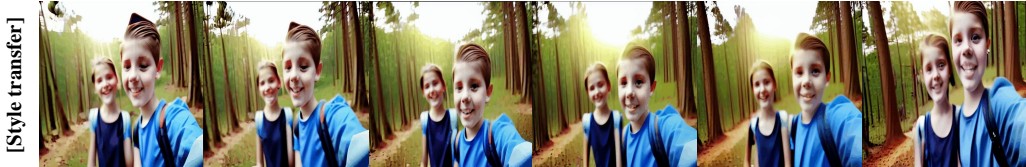

**Target Text:** *"Grandmother and granfather take selfies while walking down the street"*

[Style transfer]

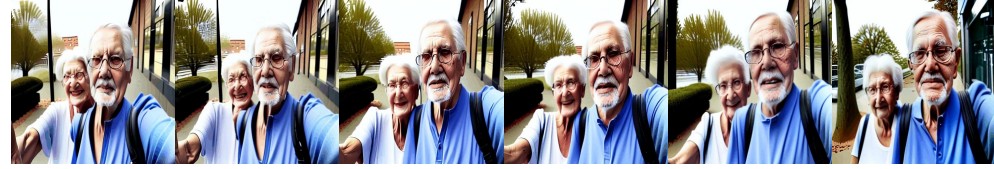

Figure 15: Edited results about style transfer.

**Input video**

**Target prompt:** *"An astronaut is skiing on the moon"*

**[Motion change]**

**Input video**

**Target Text:** *"A woman is doing yoga"*

**[Motion change]**

Figure 16: Failure case about motion change. The style of the moon is slightly changed to snow by the motion-style bias. Pre-trained model does not have much of knowledge about specific motions such as yoga.

**Limitation and Future work**    As shown in Figure 16, our studies found that some cases of editing are still challenging. We consider the following two reasons for the insufficient edits: (1) pre-trained T2I model has no knowledge of the target text and (2) certain motions (*e.g.*, skiing) in pre-trained knowledge are biased by specific styles (*e.g.*, snow) that usually come up together. Thus, our future work is to mitigate motion-style bias in editing and regularize edits to incorporate challenging edits.

**Further studies about pivotal prompt**    We performed experimental studies about the editing when a target prompt is significantly different from the contents provided in the input video. As shown in Figure 17, for a given video of a woman walking, we prepare two target prompts as (1) "An astronaut is on the moon" and (2) "goldfish in the water". The results denote that if a target prompt aligns with the content of the given video to a certain extent, it is mirrored accordingly. Conversely, when the prompt describes a completely different scenario, the model switches to generating video content based on the textual input rather than editing the existing footage.

**Input video**

**Target prompt:** *"An astronaut is on the moon"*

**Target Text:** *"A goldfish is in the water"*

Figure 17: Editing results on target prompt that has a low correlation with the original video.

