# OpenReview forum: "Pivotal Prompt Tuning for Video Dynamic Editing"
_ICLR.cc/2024/Conference — Submitted to ICLR 2024_

### Official Review · Reviewer_q1QA · 2023-10-16

**Soundness:** 3 good
**Presentation:** 3 good
**Contribution:** 3 good
**Rating:** 5
**Confidence:** 4

**Summary:**

This paper targets video editing, especially for the non-rigid, motion editing in the video. Two novel techniques are proposed.

(1) prompt pivoting tuning using a masked target prompt

(2) Spatial-temporal focusing which fuse the cross-attention map of pivotal tuning and editing

This method is compared with tune-a-video, text2video-zero in many quantitative and qualitative metrics

**Strengths:**

Provide many quantitative results in Figures 4,7 of the main paper, Table 1,2, Figure 11 of the appendix

**Weaknesses:**

My largest concern is how the results if viewed in a dense video format like mp4 or gif. For results in sparse discrete sequences in Figures, 6, 7, and 8, the difference in two consecutive is quite large.

Another point is the “dynamic motion” in the paper can be better defined. Currently, the concept of “dynamic motion” is closer to the domain of “new pose”, including “jumps”, “dance”, and “jump”. If it is a pose editing problem, a fair starting point can be a pose-driven controlnet or a text-to-pose model. If the paper targets a larger open-domain “motion”, like camera motion, fluid, and liquid motion. The better starting point is the motion prior in a pre-trained video model, because fundamentally, pre trained stable diffusion does not have knowledge about open-domain “motion”.

**Questions:**

Hope the video or long sequence results (including mp4, gif or PNG frames) can be further provided for evaluation.

Considering the time limit of rebuttal, I understand evaluation on open-domain motion can be difficult and that is not expected.

---

> ### Author Response · Authors · 2023-11-15
>
> Dear Reviewer q1QA, We address your comments and questions below.
>
> **[Q1] My largest concern is how the results if viewed in a dense video format like mp4 or gif. Hope the video or long sequence results (including mp4, gif, or PNG frames) can be further provided for evaluation.**
>
> **[A1]** Yes, we update supplementary materials with videos. Please refer to this.
>
> **[Q2] Another point is the “dynamic motion” in the paper can be better defined. Currently, the concept of “dynamic motion” is closer to the domain of “new pose”, including “jumps” and “dance”. If it is a pose editing problem, a fair starting point can be a pose-driven controlnet or a text-to-pose model. If the paper targets a larger open-domain “motion”, like camera motion, fluid, and liquid motion. The better starting point is the motion prior in a pre-trained video model, because fundamentally, pre-trained stable diffusion does not have knowledge about open-domain “motion”.**
>
> **[A2]** Yes, thank you for the attention to this matter. The dynamic motion targeted in our method is also closer to the domain of new pose. We will make this clear in the Abstract and Introduction of the main paper. (i.e., The revision is marked in blue)

---

> > ### Comment · Reviewer_q1QA · 2023-11-18
> > **Comment by Reviewer**
> >
> > Thanks for providing the video results.
> > I feel your result is quite close to Tune-a-video baselines, which is a little outdated.
> > My suggestion for improvement is that
> > (1) Since you are doing few-shot video editing, choosing a strong prior is important. I recommend the video diffusion model or Controlnet.
> > (2) The Spatial-Temporal Focusing can also be replaced with a more advanced module like FateZero.
> >
> > I would keep my previous rating

---

> ### Author Response · Authors · 2023-11-19
>
> **[Q1] I feel your result is quite close to Tune-a-video baselines, which is a little outdated.**
>
> **[A1]** With all due respect, it is absolutely false to say that our results are close to Tune-A-Video. The Tune-A-Video does not have the capability to perform non-rigid editing. It will not be able to change the motion of the object. Our method performs non-rigid editing that includes motion variation of an object, while Tune-A-Video or any other cannot. Non-rigid editing is the paper's motivation, and this paper is the first to attempt it.
>
> **[Q2] Since you are doing few-shot video editing, choosing a strong prior is important. I recommend the video diffusion model or Controlnet.**
>
> **[A2]** We are using the video diffusion model proposed by Jonathan Ho, et al, 'Video Diffusion Models’—this model is described in Section 4.1 (paragraph of `video tuning’). Our model also incorporates a strong prior with Stable Diffusion. It extends the 2D pre-trained diffusion model (i.e., Stable Diffusion) to 3D by enhancing the 2D convolution and attention layers to 3D. Furthermore, the ControlNet is not appropriate for what we want to do. Its generation is based on rigid editing given rigid guidance (e.g., canny edge, hough line). In fact, it is everything contrary to what the proposed algorithm is trying to achieve.
>
> **[Q3] The Spatial-Temporal Focusing can also be replaced with a more advanced module like FateZero.**
>
> **[A3]** Our method covers a more general module of attention control than FateZero catered for video and performs the dynamic motion variation that we are trying to achieve. The module used in FateZero is the cross-attention control module in the work proposed by Hertz, Amir, et al, ‘Prompt-to-Prompt’ formulated as ( $x = \alpha_w \odot x + \beta_w \odot y$ ). Our proposed cross-attention control (i.e., Spatial-Temporal Focusing) ( $x^{i} = \gamma^{i} (\alpha_w \odot x^{i} ) + (1−\gamma^{i})(\beta_w \odot y^{i})$ - given in page 7 of the manuscript) allows the capability to perform frame-level control by leveraging temporal weight $\gamma^{i}$ for each $i$-th frame, which enhances temporal dynamic editing along the frames.
>
> ($\alpha_w, \beta_w$ are coefficients for blending, $x$ is the cross attention map for editing and $y$ is the cross attention map for tuning process. $\odot$ is element-wise multiplication with broadcasting.)

---

> ### Comment · Reviewer_q1QA · 2023-11-19
> **Official Comment by Reviwers**
>
> Thanks for pointing out the difference.
> I get the difference between your method with Tune-a-video, and the controller.
>
> Anyway, I would like to share the results of Tune-a-Video + ControlNet here.
> First, tune the stable diffusion model on the input video as
> https://tuneavideo.github.io/assets/data/man-basketball.gif
> Then, they prepare the skeleton of the new motion which can be different from the input video motion
> https://tuneavideo.github.io/assets/results/tuneavideo/pose/man-basketball/dancing/pose.gif
> https://tuneavideo.github.io/assets/results/tuneavideo/pose/man-basketball/running/pose.gif
> As we can see, the final result is also a ``non-rigid editing'' here:
> https://tuneavideo.github.io/assets/results/tuneavideo/pose/man-basketball/dancing/video.gif
> https://tuneavideo.github.io/assets/results/tuneavideo/pose/man-basketball/running/video.gif

---

> ### Author Response · Authors · 2023-11-20
>
> Thank you for sharing the video. Upon viewing the video and double checking with the GitHub of Tune-a-video, it's evident that this is not an example of non-rigid editing. Instead, the dancing video appears to have been previously prepared using a skeleton-based video followed by rigid editing based style transfer.

---

### Official Review · Reviewer_9TJK · 2023-10-27

**Soundness:** 3 good
**Presentation:** 3 good
**Contribution:** 3 good
**Rating:** 5
**Confidence:** 4

**Summary:**

The paper proposed a Pivotal Dynamic Editing (PDEdit) framework for text-based video editing task.  PDEdit allows non-rigid edits to a general video using the target text. PDEdit is versatile and effective for editing various types of videos. Extensive experimental results reveal the effectiveness of the proposed method.

**Strengths:**

1. A versatile video editing method that allows non-rigid editing is proposed.

2. The motivation is clear, and experimental results reveal the effectiveness of the proposed method.

**Weaknesses:**

1. The concept of "non-rigid edit" is difficult to understand, existing methods (e.g., Tune-A-Video) can edit the input video so that the motion is inconsistent with the original video, isn't this non-rigid editing?

2. The proposed PDEdit involves many hyperparameters to achieve good results, and requires additional Charades-STA dataset, which makes the practical application process full of challenges.

3. Since the prompt corresponding to the original video is not used, if the target prompt has a low correlation with the original video, will this cause the first tuning stage to fail?

4. Some "rigid edit" methods such as Gen-1, VideoComposer and Control-A-Video, which incorporate structure guidance from input videos to generate videos, should also be discussed in the related work.

5. Lack of comparison with existing open-source editing methods such as Pix2video and Video-p2p.

6. Authors should include edited sample videos in supplementary materials or on anonymous websites for easy comparison.

**Questions:**

See Weaknesses for more details.

---

> ### Author Response · Authors · 2023-11-15
>
> Dear Reviewer 9TJK, We address your comments and questions below.
>
> **[Q1] The concept of "non-rigid edit" is difficult to understand, existing methods (e.g., Tune-A-Video) can edit the input video so that the motion is inconsistent with the original video, isn't this non-rigid editing?**
>
> **[A1]** Non-rigid editing involves synthesizing the shape of content to align with a specified target prompt. In contrast, rigid editing focuses on altering the style of content while preserving its shape. Existing methods have performed rigid-editing such as inpainting. As shown in Figure 2 of the main paper and Appendix B, our studies found that current editing systems (e.g., Tune-A-Video) lack the capability to perform non-rigid editing such as motion editing.
>
> **[Q2] The proposed PDEdit involves many hyperparameters to achieve good results and requires an additional Charades-STA dataset, which makes the practical application process full of challenges.**
>
> **[A2]** PDEdit takes a single hyperparameter about deterministic score s*, where Charades-STA dataset is used for investigating the optimal value of s*. In an effort to alleviate the reliance on additional datasets, Figure 11 in the Appendix conducts a sensitivity analysis on s*, establishing a robust operating range between 20 and 22. This ensures the proper functioning of the system without necessitating additional datasets for the reproduction of s*.
>
> **[Q3] Since the prompt corresponding to the original video is not used, if the target prompt has a low correlation with the original video, will this cause the first tuning stage to fail?**
>
> **[A3]** To identify this, we perform an experiment on the video in Figure 6 of the main paper (i.e., a video of a woman walking). We prepared two target prompts as (1) 'astronaut on the moon' and (2) 'goldfish in the water' The results are updated in Figure 17 of the Appendix. If a target prompt describes a situation that can be applicable to the given video to a certain degree, it is mirrored accordingly. However, if the prompt describes an entirely different scenario, the model switches to generating video based on the text rather than editing. Thank you for the attention to this matter. We will further elaborate on these findings.
>
> **[Q4] Some "rigid edit" methods such as Gen-1, VideoComposer, and Control-A-Video, which incorporate structure guidance from input videos to generate videos, should also be discussed in the related work.**
>
> **[A4]** Yes we add them in the revision, please check our updated version. (i.e., The revision is marked in blue)
>
> **[Q5] Lack of comparison with existing open-source editing methods such as Pix2video and Video-p2p.**
>
> **[A5]** We include the reuslts of Video-P2P and Pix2Video into our current validation in Table 1. We assess the non-rigid/rigid editing capabilities of current editing systems with respect to textual alignment, frame consistency, and fidelity to the input video. Textual alignment measures the average CLIP score between the target prompt and the edited video frame. For frame consistency, we consider both the average CLIP score among sequential frames and the FVD between input and output videos. To evaluate fidelity to the input videos, we mask the edited regions in the edited videos and measure PSNR, LPIPS, and SSIM scores. PDEdit exhibits superior performance compared to previous editing systems, particularly demonstrating significant improvements in non-rigid editing. We also update these in Table 1 of the Appendix.
>
> **[Q6] Authors should include edited sample videos in supplementary materials or on anonymous websites for easy comparison.**
>
> **[A6]** Yes, we update supplementary materials with videos. Please refer to it.
>
> ***Table Specifications***
>
> **Textual Alignment**: Clip(text-video)
>
> **Temporal Consistency**: Clip(image-image), FVD
>
> **Fidelity**: PSNR, LPIPS, SSIM on the unedited area (i.e., applying mask on edited area)
>
> **A/B**: non-rigid edit/rigid edit
>
> **Table 1**
>
> ||Clip(text-video) ↑|Clip(image-image) ↑|FVD ↓|PSNR ↑|LPIPS ↓|SSIM ↑|
> |:---|:---:|:---:|:---:|:---:|:---:|:---:|
> |PDEdit|26.1/27.9|93.1/94.2|2831/2521|19.75/21.24|0.3672/0.3121|0.7135/0.7821|
> |TAV|16.4/26.0|89.8/92.6|3403/2720|14.62/17.46|0.5776/0.4452|0.5425/0.6271|
> |T2V-Zero|13.7/24.9|86.1/87.4|4052/4235|9.31/11.73|0.5902/0.5732|0.4090/0.4264|
> |Video-P2P|15.1/25.8|91.6/93.4|3261/2683|16.17/18.46|0.4502/0.3951|0.5832/0.7183|
> |Pix2Video|15.9/25.8|90.4/91.8|3131/2704|16.09/18.31|0.4964/0.4211|0.5609/0.7293|

---

### Official Review · Reviewer_UHtg · 2023-10-27

**Soundness:** 2 fair
**Presentation:** 2 fair
**Contribution:** 2 fair
**Rating:** 5
**Confidence:** 4

**Summary:**

This paper focuses on spatial-temporal non-rigid video editing, a relatively underexplored area in video editing tasks. The proposed method, Pivotal Dynamic Editing (PDEdit), distinguishes itself by not requiring original video captions but editing videos based on suggested prompt pivoting. The method demonstrates a degree of generality.

**Strengths:**

- Unlike previous video editing approaches, which mainly focus on simple edits like appearance and style, this paper pioneers motion editing, showing a forward shift in research focus.
- The method proposed here is innovative, deviating significantly from existing pipelines.
- The writing is clear and relatively easy to follow.

**Weaknesses:**

- The visual results presented in the paper are subpar. While the difficulty of the task is understood, the demonstrated demos are unsatisfactory in their current state.
- The experimental section is weak, lacking explicit numerical comparisons in the main text. The experiments are limited to only 20 videos, which is a small sample size. The comparison metrics lack persuasiveness.
- Although supplementary materials include editing demos, the videos appear to lack continuity, and there seems to be a limitation in supporting video editing across different resolutions.

**Questions:**

See above.
- I am uncertain about how many frames of video the author's method supports editing. It appears to be limited to very short 8-frame videos, which have limited practical significance.
- Stacking images as a presentation method is not ideal; I would prefer actual demos in GIF or video format.
- Despite these shortcomings, I recognize the contribution of the authors to this underexplored area. Their method might inspire future research. However, at the current stage, I can only give a borderline score.

---

> ### Author Response · Authors · 2023-11-15
>
> Dear Reviewer UHtg, We address your comments and questions below.
>
> **[Q1,3] The visual results presented in the paper are subpar. While the difficulty of the task is understood, the demonstrated demos are unsatisfactory in their current state. Although supplementary materials include editing demos, the videos appear to lack continuity, and there seems to be a limitation in supporting video editing across different resolutions.**
>
> **[A1,3]** To make this clear, we updated supplementary material with videos. Please refer to it.
>
> **[Q2] The experimental section is weak, lacking explicit numerical comparisons in the main text. The experiments are limited to only 20 videos, which is a small sample size. The comparison metrics lack persuasiveness.**
>
> **[A2]** We conduct experiments involving numerical comparisons, as detailed in Table 1 of the Appendix. Our evaluation encompasses textual alignment (CLIP), frame consistency (CLIP, FVD), and fidelity to the input video (PSNR, LPIPS, SSIM). Additionally, in Table 2 of the Appendix, we extend our validation to include more samples from other video datasets, (i.e., UCF101, WebVid-10M). We will consider to incorporate these into the main paper.
>
> **[Q4] I am uncertain about how many frames of video the author's method supports editing. It appears to be limited to very short 8-frame videos, which have limited practical significance.**
>
> **[A4]** Considering our GPU resource (i.e., A100 GPU), our experiments are conducted on video sequences spanning 24 to 40 frames. Our proposed method does not rely on the length of the video.
>
> **[Q5] Stacking images as a presentation method is not ideal; I would prefer actual demos in GIF or video format.**
>
> **[A5]** To make this clear, we add supplementary results with video. Please refer to the updated materials.
>
> **[Q6] Despite these shortcomings, I recognize the contribution of the authors to this underexplored area. Their method might inspire future research. However, at the current stage, I can only give a borderline score.**
>
> **[A6]** Thank you for the recognition. Of course, we can not do all of it. But, to the best of our knowledge, PDEdit represents a pioneering effort in tackling non-rigid editing, particularly in the realm of motion editing. We aspire for our work to make meaningful contributions to these research domains. We ensure to release our code and project.

---

### Official Review · Reviewer_1N5p · 2023-11-01

**Soundness:** 1 poor
**Presentation:** 2 fair
**Contribution:** 2 fair
**Rating:** 5
**Confidence:** 3

**Summary:**

This paper introduces a method for text-based video dynamic editing, which involves making non-rigid spatial-temporal alterations.  They propose pivotal prompt tuning for the system to tune a video with target text prompt and temporal dynamic editing to apply motion changes in spatial and temporal domain.

**Strengths:**

1. This paper proposes a novel approach to text-based video dynamic editing
2. The video frames shown in the paper demonstrate edited motion similar to the text-prompt inputs.

**Weaknesses:**

1. The absence of accompanying videos in a video editing paper makes it challenging to assess the temporal consistency of the edited frames.
2. The quality of the editing falls short of expectations, with significant color discrepancies and noticeable object alterations post-editing. It seems like the unedited area is not preserved after editing for the applications outside of style transfer.
3. The quantitative evaluation should also evaluate how this editing method preserves the unedited area. For example, include PSNR, SSIM or LPIPS for the unedited background area for motion editing.

**Questions:**

1. In distributional pivoting, do you need to tune the parameter for s* for different videos?

**Details Of Ethics Concerns:**

Video motion editing could be abused for illegal behavior that should be concerned.

---

> ### Author Response · Authors · 2023-11-15
>
> Dear Reviewer 1N5p, We address your comments and questions below.
>
> **[Q1] The absence of accompanying videos in a video editing paper makes it challenging to assess the temporal consistency of the edited frames.**
>
> **[A1]** We uploaded a supplementary material containing the videos.
>
> **[Q2] The quality of the editing falls short of expectations, with significant color discrepancies and noticeable object alterations post-editing. It seems like the unedited area is not preserved after editing for the applications outside of style transfer.**
>
> **[A2]** There is a variance in the quality of editing results due to the text-scene bias of the pre-trained diffusion model (e.g., Stable Diffusion). For example, as shown in Figure 16 about Limitation in the Appendix, we edit a walking astronaut’s motion as skiing, and the results show the man skiing on a snowy scene. This suggests that the editing about skiing exhibits a bias towards scenes involving skiing on snow. Consequently, the quality of editing depends on whether the required editing by the target prompt is bias-aligned or bias-conflicted.
>
> **[Q3] The quantitative evaluation should also evaluate how this editing method preserves the unedited area. For example, include PSNR, SSIM, or LPIPS for the unedited background area for motion editing.**
>
> **[A3]** Yes we did. In Table 1 of the Appendix, we evaluated PSNR, SSIM, and LPIPS for the unedited area by applying the mask to the edited region.
>
> **[Q4] In distributional pivoting, do you need to tune the parameter for** s* **for different videos?**
>
> **[A4]** Yes, the s* is a trainable parameter determined under the training video dataset. In Figure 11 of the Appendix, our investigation delves into the sensitivity analysis of "s*." The findings reveal a stable working region defined by 20 < s* < 22. This implies we can leverage this specific range for optimal score selection without training videos.
>
> **[Q5] Video motion editing could be abused for illegal behavior, and that should be a concern.**
>
> **[A5]** Within our main paper's ethical statements, we acknowledge and align with concerns regarding the illicit use of video editing. Consequently, we explore viable solutions, such as forensic analysis and digital watermarking, to mitigate these ethical issues effectively.

---

> > ### Comment · Reviewer_1N5p · 2023-11-22
> >
> > Thanks for the video provided. From the video, I cannot find consistent editing even for unedited areas. Although the proposed method uses stable diffusion which makes the editing inconsistent, there are some other works of video generation that produce consistent background, for example, Emu video. Therefore, I'll keep my original rating

---

### Author Response · Authors · 2023-11-15

Dear Reviewers,

We included supplementary material about the video. These results are intended to enhance our responses to the valuable comments provided by the Reviewers.

---

### Meta-Review · Area_Chair_Jeez · 2023-12-07

**Metareview:**

This paper proposes a novel approach for text-based video dynamic editing, involving non-rigid spatial-temporal alterations. The method puts forward two innovative techniques: prompt pivoting tuning, achieved through a masked target prompt, and spatial-temporal focusing, which integrates the cross-attention map of pivotal tuning and editing. A distinctive feature of the proposed method is its independence from original video captions, relying instead on the editing of videos through suggested prompt pivoting.

While prevalent methods primarily concentrate on modifying appearance and style, the problem of text-based motion editing remains a novel and underexplored challenge. The proposed method is innovative, markedly diverging from established pipelines. The visual results showcased in the paper illustrate motion editing that closely aligns with the text-prompt inputs.

The primary concern is the quality of the edited results. Several issues contribute to this concern, including notable color discrepancies, noticeable object alterations, changes in the unedited area, and a lack of temporal continuity. Additionally, the experiments are limited in size. While acknowledging the inherent difficulty of the task, the current state of the demonstrated results is deemed unsatisfactory and falls short of meeting the standards set by ICLR. Minor issues, such as metrics lacking persuasiveness, hyperparameter tuning, and comparisons with existing open-source editing methods, are partially addressed. However, the consensus among reviews suggests that the overall quality still fails to meet the standard for ICLR.

**Justification For Why Not Higher Score:**

While the problem and approach are novel, the quality of the editing results does not meet the standards of ICLR. The issues include perceivable color discrepancies, noticeable object changes, alternations in the unedited area, and a lack of temporal continuity.

**Justification For Why Not Lower Score:**

N/A

---

### Decision · Program_Chairs · 2024-01-16

Reject